# Percutaneous Discectomy Followed by CESI Might Improve Neurological Disorder of Drop Foot Patients Due to Chronic LDH

**DOI:** 10.3390/brainsci10080539

**Published:** 2020-08-11

**Authors:** Trianggoro Budisulistyo, Firmansyah Atmaja

**Affiliations:** 1Department of Neurology, Diponegoro Univ/Dr.Kariadi Hospital, Semarang 50244, Indonesia; 2Indonesian Army Health Center/Pelamonia Hospital, Makassar 90157, Indonesia; padmaatmaja@gmail.com

**Keywords:** LDH, drop foot, fluoroscopic, discectomy, CESI

## Abstract

(1) Introduction: Epiconus and conus medullary syndromes that consisted of drop foot, pain, numbness, bladder or bowel dysfunction are serious problems might be caused by lumbar disc(s) herniation (LDH) compression. (2) Objective: To evaluate percutaneous discectomy effectivity for decompressing LDH lesions. (3) Case Report: Three patients suffered from drop feet, numbness, and bowel and bladder problems due to LDH compression. Patient #1 is a male (35 years old, basal metabolism index (BMI) = 23.9), point 1 on manual muscle test (MMT), with protrusion on L3 to S1 discs; Patient #2 is a female (62 years old, BMI = 22.4), point 3 on MMT, with protrusion on L2-4 and L5-S1 discs; Patient #3 is a female (43 years old, BMI = 26.6), point 4 on MMT, with extrusion on T12-L1 and L1-2 and L3-4 protruded discs. Six months follow-up showed of stand and walkability improvement with Patient #1 and #2. Patient #3 showed improvement in bowel and bladder problems within 10 weeks, without suffering of postoperative pain syndromes. (4) Discussion: Patient #1 and #2 showed better outcomes than Patient #3 who affected epiconus and cauda equina syndromes. Triamcinolone and lidocaine have analgesic and anti-inflammatory properties for improving intraepidural circulation adjacent to the lesion sites. (5) Conclusion: Drop foot caused by mechanical compression of LDH ought to be treated immediately. Lateral or posterolateral compression has better outcomes associated with anatomical structures. Discectomy through transforaminal approach that is followed by caudal epidural steroid injection (CESI) under fluoroscopic guidance is a safer and minimally invasive treatment with promising outcomes.

## 1. Introduction

Drop foot is commonly originated from lower motor neuron lesions thus can be classified as radiculopathy or peripheral neuropathy. Approximately, 68% of cases are caused by peripheral neuropathy, such as polyneuropathy (18.3%), L5 radiculopathies (19.7%), and peroneal neuropathy (31%). Its account for, approximately, 15% of all mononeuropathies; in adults, it is caused by nerve entrapment of lower extremities. Presence of numbness or pain associated to body position or dermatomal distribution may be due to nerve roots disorder called radiculopathy. Drop foot syndromes can be caused by L4-5 (20%) or L5-S1 (41.5%) or isolated L5 (25.2%) nerve roots compression, and thus might lead to degenerative lumbar disease. Patients with L5 radiculopathy may present dorsiflexion palsy as well as diminished reflexes of knee (PTR) and ankle (ATR). Positive result of hip abduction can be seen in lumbar radiculopathy in 85.7% of drop feet patients, whereas peroneal neuropathy is seen in only 3.6% patient [1].

Lumbar disc(s) herniation (LDH) and stenosis are degenerative disorders underlying chronic low back pain (cLBP) but rarely cause with drop foot syndrome [2,3]. L4-5 disc level is the common lesion site for the occurrence of LDH, therefore, dropping foot would be cause by L5 root compression (innervate tibialis anterior and hallucis longus muscles). However, the L4 and S1 roots also innervate tibialis anterior and hallucis longus muscles based on electrical stimulation examination [2]. Two years postoperative follow-up of drop foot patients showed significantly better outcomes on early (79.1 + 21.2%) than delayed decompression groups (68.6 + 18.6%) [4]. Analgesics medication is still prescribed routinely in common cases of CLBP patients for relieving the neuropathic pain syndrome by neurotransmitter modulation at presynaptic receptors of afferent fibers [5]. This case report is purposed to assess the neurological improvement in drop foot patients due to LDH that underwent percutaneous discectomy and CESI treatment.

## 2. Procedure Technique

The author, a neurologist, and pain consultant are trained for the minimal invasive treatment of the spine and interventional pain management. The author has experience of doing percutaneous discectomy on more than 150 patients (since 2017), to LDH or stenosis, and interventional pain management on more than 1500 cases (since 2012), defined as chronic or intractable pain syndromes. All patients have been informed of the purpose, benefits, and potential complications or the side effect of the study, and they have signed the informed consent. The potential complications are worsening of the motor, sensory abnormalities, bowel and bladder dysfunction, persistent pain, and hypersensitive reactions to local anesthetic (lidocaine), dye contrast agent, antibiotic, the discectomy kits, or suture material. Percutaneous discectomy procedure only requires a local anesthetic administration, so the patients are still awake and able to communicate with the operator when the needle or dissection procedures touch the nerve root or when they have other inconveniences, to minimize the potential complication. Moreover, during the procedure, the operator can check the patient’s motor or sensory function.

During the surgery, the patients were given an intravenous line of normal saline, and they were connected to either the vital sign or pulse oximetry monitors. Since patients lied in a prone position aseptic or antiseptic and scrubbing underwent on the target level(s). Skin markings were done by following the rule of process line and disc(s) line in the PA view and AP line in lateral view. These intersected lines served as the point of entry for the procedure. The procedure started by injecting 1.5% of lidocaine then followed by 18G of spinal needle insertion at once injected up until disc surface. When the needle was fixed, then C-arm fluoroscope was turned in lateral view for monitoring and advancing the insertion and was stopped at one-third of the posterior part of the discs. Precisely, 1 mL of dye contrast was injected inside the disc to observe annular tears or contained structure (discography). C-arm fluoroscope was turned again to the PA view to make sure that the tip of the needle did not cross the midline (processes line), and needle’s stylet was removed. Then, the guidewire was inserted to replace it. One centimeter of a small incision was done at the point of entry for inserting the minidisc tube, which was carefully inserted following the guidewire direction. When it reaches neuroforamen doom, the adjacent tissues might get stuck, so it should be properly maneuvered. Gentle hammering was done for advancing the tube insertion under C-arm fluoroscopy guided by guidewire until it reached to posterior one-third of disc. A mini grasper was used to evacuate 0.8 g or 2 mL of protruded disc as figuring on the MRI. We were concerning to catch the “tail” of protruded disc, so might be able to remove much more of the outsider parts. Shortly, it was followed by bipolar coagulation system of nociceptors within the posterior part of the disc. One gram of cefazoline was injected inside the disc (2–3 mL), then the tube was pulled out and the skin incision was sutured after completion of the procedure. In principle, this percutaneous discectomy has similarities with transforaminal micro-discectomy (TFMD) technique, but without endoscopic kits.

The CESI procedure was done by firstly inserting a needle through sacral hiatus under C-arm fluoroscopy in the midline direction and stopping when it reached below the S2 or 3 levels. The needle placement was confirmed by using dye contrast (iopamidol) resembling a Christmas Tree in a C-arm fluoroscopy monitor. Second, triamcinolone 10 mg dissolved with 1% lidocaine and 1.5 mL of normal saline to a total volume up to 8 mL was injected gently through the inserted needle on sacral epidural space. Finally, the patients were brought into the recovery room and monitored for at least 30 min for any complications (Figure 1). 

## 3. Case Report

This article reports three patients with similar main complaints of chronic lower back pain or radicular pain as NRS equal to 3–4, drop foot, numbness on lower limbs, and bowel and bladder dysfunction. All patients had done MRI examinations with definite diagnoses of discs herniation. They have been following for all treatment modalities such as medication and physical rehabilitation without any satisfactory improvement. Patient #1 (C 784421) is a male (35 years old, basal metabolism index (BMI) = 23.9), patient #2 (C 7555240) is a female (62 years old, BMI = 22.4), and patient #3 (C 804357) is a female (43 years old, BMI = 26.6). They had been hospitalized and underwent minimally invasive spine treatment at Dr. Kariadi Hospital in Semarang, Indonesia. Patient #1, a construction field worker, has a repeated microtrauma injury that occurred more than 2 years ago. He suddenly suffered lower back pain during his work and had dropped feet as showed by point 1 on the MMT scale, with positive PTR, reduced ATR, numbness, and bowel and bladder dysfunction. MRI result showed posterocentral protruded discs on L3-4, L4-5, and L5-S1 level, and posterolateral bulged disc compression on L2-3. Patient #2 has a stroke syndrome and got medication in 2018 following a traumatic fall down accident on an airport elevator. She had a dropped foot by the motor function registered at 3 points on MMT scale, numbness, and bladder and bowel dysfunction. MRI results showed protruding disc on both lateral recesses of L3-4 levels causing central canal stenosis and disc bulge on the L5-S1 level. She had undergone physical treatments and medication previously, and in October 2019, she underwent percutaneous discectomy. Patient #3 is a female (43 years old) with weakness on calves (4 points on MMT scale), dropped feet, numbness around the lower back, buttock, and feet, bowel and bladder dysfunction, and decreasing of PTR and negative value of ATR examination caused by motorcycle accident at November 2018. MRI findings showed either an extruded disc, on T12-L1 level, protrusion on L1-2, and L3-4 levels, or stenosis causing posterolateral and central compression.

Postoperative oral prescription consists of levofloxacin (500 mg/day), gabapentin (150–200 mg/day), acetaminophen (1000–1500 mg/day), thiamine (100 mg/day), calcium lactate (1000 mg/day), and sodium diclofenac (100 mg/day) for 1 week duration. They asked for visiting back to the Dr. Kariadi Hospital in Semarang after 1 week. Postoperative monitoring is aimed to take care of infectious complication, delayed hypersensitive reaction, pain scale, neurological examination, and wound healing. They have slightly improvement in numbness without serious complications. Pain scales on mild spectrum in accordance to operative sites. This can be treated with gabapentin (150–200 mg/day) and acetaminophen (1000–1500 mg/day), wherein acetaminophen is only given for moderate level pain. The antibiotic, levofloxacin 500 mg/day was only taken for 7 days postoperation. Postoperative evaluation at month 6 showed increasing of 2 points on the MT scale with betterment of standing and walking in a short distance for patient #1. Patient #2 showed, for lower limb, increasing of 1 point on the MMT scale and assisted walking for 10–15 m of distances. Patient #3 did not show advancement in motor function and drop foot, or the numbness on foot area, has settled at week 10, but the bowel and bladder function showed improvement. They did not have significant postoperative pain syndromes including radicular or discogenic pain (Figure 2 and Figure 3).

## 4. Discussion

Drop foot syndrome is defined as muscular paresis or paralysis with underlying few etiologies and also in stenosis or herniated discs might cause lesions at nerve roots or centrally to the cord. Compression of the L4, L5, or S1 roots might cause the laxity of tibialis anterior muscle and is shown as a drop or scrubbed foot when walking. Lateral compression on the disc is according to neurologic deficits than pain complaints [6]. Decompression or surgical treatment applied to drop foot patients due to degenerative disorders has shown recovery ranging from 61% to 88%, while no recovery of 28.3%. Influencing factors resulting in an improved outcome are the age of fewer than 65 years old, duration of onset less than 6 weeks, or preoperative without any severe syndromes. Obesity seems to have no significant effect. Operation procedures were benefited from decompressing purposes but not to inflamed nerve roots. Hence, the administration of anti-inflammatory agent can optimize the results. Reversal of foot drop depends on the time for the nerve to recover due to anti-inflammatory agents or compressive lesions. The significant recovery of muscle strength has been reported up to 6 months postoperation, but it depends on the age and duration or gradation of weakness [2,7].

Compressing root on L5 and S1 level are the most common findings (41.5%), dominated by L5 roots (93.3%) and less by L4 or S1 [8]. Mechanical compression by a huge amount of discs prolapse or stenosis either uni- or bilaterally might lead to radicular pain, bowel and bladder dysfunction, sensory and motor problems either to drop foot [6]. Decompression treatment officially showed motor improvement of more than 4 points on MMT scale for 15% cases, whereas 60–70% cases were below 4 points on MMT. The improvement level was influenced by the procedure or underlying etiologies, term duration of onset, and severity of motor weakness. Delayed time for surgery results in drop foot and decreasing the probability of improvement by 33%. In 2016, Chun and Park had undergone PELD to L5–S1 far lateral disc herniation associated with drop foot. They have shown a significant improvement of pain and a full recovery of foot drop afterward and similarly with Wang et al. [8]. Thoracic vertebral column patterned by kyphosis on the upper level, lordosis, and kyphosis at the thoracolumbar junction (T12–L1) was seen. The prevalence of thoracic disc herniation caused by a traumatic accident is still rare but it comprises 1% of total disc herniations. Few studies mentioned that the incidence of thoracic disc herniation is relatively high in lower level and thoracolumbar junctions of 75% at T8 to L1 level (Figure 4).

The cord lying along the thoracic and thoracolumbar level is vulnerable with little buffer space of the canal. Many of large arteries or venous structures, such as the Adamkiewicz artery on the left side, may be damaged. Pulmonary or neurological complications risks can be minimized by doing lateral or posterolateral approaches through neuroforamen (transforaminal approach). Immediate decompression procedures of thoracic or upper LDH in the acute phase might result in 60–80% of complete improvement, 27% of partial improvement, and 12% without any changes [7,9,10]. Complete improvement was found in 61% of patients with drop foot syndrome whose onset duration was less than 4 weeks [7].

Patient #1 suffered from dropped foot suddenly when doing his job in the construction field for the entire 2 years. He got repetitive microtraumatic injury for the last 2 years. While at work, he suddenly got lower back pain and feet dropped as the motor function showed 1 point on MMT scale, and PTR was positive but the ATR was reduced, with numbness and bowel and bladder dysfunction. The MRI result showed protruded discs on L3-4, L4-5, and L5-S1 levels posterocentral and posterolateral and bulged disc on L2-3 level. Patient #2, besides suffering from a drop foot caused by a traumatic accident, also experienced stroke syndromes. She was prescribed for stroke medication in 2018 and had a traumatic accident having a fall on an airport elevator in the same month. Afterward, she had lower limb weakness and dropped feet of point 3 on MMT scale, numbness, and bladder and bowel problems. MRI result showed protruding disc on both lateral recesses of L2-3 and L3-4 levels, cause of central canal stenosis, and disc bulging on the L5-S1 level. They both had undergone physical treatments and medication previously. In October 2019, they had undergone percutaneous discectomy, and 10 mg of triamcinolone mixed with 1% of lidocaine and 1.5 mL of normal saline was injected intraepidurally. Clinical findings of pain and/or numbness were done on L2 (on thigh area), L3 (on the medial aspect of knee joint), L4-5 (on the lateral aspect of the thigh to calves), and S1 (on heel and foot thumb) levels. Kido et al. suggested that disturbance of L2 root might have a normal tibialis anterior muscle function (point 5 on MMT scale) and the PTR examination has depressed or disappeared in L4 root disorder. This syndrome is often differed for each individual, hence the radiographic examination such as MRI and CTM have been used to define it [11]. Patient #3 was having 3 sites of the lesion, including extruded discs on T12-L1 and L3-4 levels known as thoracolumbar junction or upper LDH and stenosis of the spinal canal. It was found that around 30% of these sites’ lesions tend to present epiconus or cauda equina syndrome. A recent study found that only 58% and 53% of patients, respectively, in the L1-L2 and L2-L3 group with an improvement of radicular and back pain underwent decompression by fusion surgery. A long-term follow-up study confirmed a worsening outcome of the disc lesion on L1-L2 and L2-L3 levels, of which only 33% of patients reported a better result in economic or functional status. The postoperative outcome in L3-L4 group might achieve improvement as approximately 87–94% [12,13]. Thus, delayed time of decompression treatment could reduce the possibilities of better outcomes [10,14,15].

The age of fewer than 45 years old and the duration of onset of fewer than 6 months might result in better outcomes. The zone of disc herniation influences the outcome as the posterolateral approach could have an excellent outcome (57.6–61.5%), followed by the foraminal (42.9%) and posterocentral (15.4%) lesions [16]. It was similar to patients #1 and #2 that showed improvement on MMT scale of at least 1 point, and bladder and bowel dysfunction got quite better gradually than it was 6 months ago. Patient #3 has gradually improved on bowel and bladder problems but not to the motor weakness. It was associated with the high severity of preoperative symptoms, multiple sites of herniated disc lesion, and long-term delayed decompressing treatment [15,17]. Patient #1 has better motor outcomes gradually at 6 months, even though he had posterocentral protrusion on L3-4, L4-5, and L5-S1 levels. His dorsiflexion ability of foot was not improved enough although his walking function was quite better and promising going forward. Clinical improvement on patient #2, who aged older than 45 years, seem related to the traumatic mechanism as she fell on an elevator in 2018. It could not be classified as a severe traumatic accident, although the MRI figuring protruded disc on both lateral recesses of L2-3 and L3-4 levels and bulging disc on L5-S1 level. Patient #3 suffered from multiple sites and severity stage of the lesion related to the outcome. Upper lumbar lesions that were indicated by extremely advanced compression on T12-L1 level and protruded discs on L1-2 level were significantly associated with epiconus syndromes. Those sites of herniated disc lesion are epiconus (T12-L1) and conus medullary (L1-2 and L3-4). Extreme lesions showed on T12-L1 and L3-4 are called extrusion discs and stenosis of the spinal canal, respectively. It is compressed and affects the lowest part of the spinal cord, even the L1-2 sites called as a protruded disc herniation, and conus medullary of the spinal cord [17,18].

The percutaneous discectomy procedures are based on the understanding of foraminal and extraforaminal structures, Kambin’s triangle, and disc zones as figured by MRI. Based on the anatomical structures, hence transforaminal approaches seem safer and more effective for them. This procedure is quite safe because the working zone is the one-third posterior part of the discs, it avoids removal of annular layers, and using a small caliber of disc tube and grasper, a small amount of material (5 mL or the same as 0.8 g) is removed. It can be used not only for lateral, paramedian foraminal herniation but also for removing disc material in the midline by proper experienced skill. For each removal time, the operator should check the grasper’s tip under C-arm fluoroscopy guidance. Bipolar coagulation is aimed to complete the remaining nucleus pulposus by inserting a device into the disc and performing circumferentially. This coagulation might seal the microbleeding inside the disc and prevent further reherniation [19]. Patients were discharged 24 h after and given postoperative care instructions without any additional bracing. They should be careful about doing activities such as weighted lifting, squat position, long-term sitting, especially on the couch, and sleeping (sleep position should be parallel between shoulder to lower back) for approximately further 3 months. A conventional open surgery occasionally requires manipulating the structures adjacent to neuroforamen, thus increasing the risk of instability, but not to transforaminal approach. The thoracic disc herniation that developed on the traumatic basis will be softer than degenerative processes, so it could be possible to treat it with posterolateral approaches. Limitations of this transforaminal approach are in L5–S1 level lesion, herniated disc caudal migration, or even in cauda equina syndrome. By doing this transforaminal percutaneous discectomy under C-arm fluoroscopy guiding might avoid the risk of nerve or vessel injury, dural puncture, muscle injury, or instability of lamina and facet joints. Besides it can minimize the cost of therapy as the hospital LOS might reduce and the patient can be mobilized earlier [9,15,20]. All of the patients have done postoperative CESI that aimed to treat the inflammation and postsurgical pain, by inhibiting the enzyme phospholipase A2 which hydrolyzed the bond converting membrane phospholipids into arachidonic acid and lysophospholipids. The phospholipase A2 is a kind of an inflammatory mediator whose concentration might be elevated in degenerative or disc herniation. Other than that, steroids administration might inhibit pain pathways by suppressing ectopic discharges from injured nerves and lowering the conduction velocity of unmyelinated C fibers. CESI is the safer approach for postsurgical patients than other methods for avoiding dural or vascular puncture and can reach the ventral part of epidural space. It has good (level I) evidence in the back and radicular pain due to disc herniation and discogenic pain [21]. The CESI administration in this case reports was done by mixing triamcinolone 10 mg, lidocaine 1%, and 1.5 mL of 0.9% normal saline (total injected volume was 8 mL). The purpose of injecting 2 mL of normal saline afterward is to flush out the steroids and wash out inflammatory cytokines inside the epidural space. Triamcinolone is a particulate steroid that theoretically might cause complications, such as vascular penetration, dizziness, nausea, an increase of blood sugar or tension, arrythmia, or even paraplegia due to nerve damage. However, it did not happen among them [21,22]. There is no correlation between the amount of drug dose and the intraepidural therapeutic effect of steroids for reducing pain or neurological improvement. Administration of intraepidural 0.9% normal saline after CESI might help to flush and wash out the remaining steroid, inside the needle, and inflammation cytokines, in epidural space. Few clinical studies of CESI have demonstrated better results for LDH than spinal stenosis [21]. At least 70% of patients who had ESI showed improvement of radiculopathy within 6 months and gradual resorption of the herniated disc within 1 year of presentation. Or whether the herniated disc size remains, and the modulation of inflammatory response have improvement [23,24]. Intraepidurally administered local anesthetic drugs can improve circulation adjacent to the compressing nerve roots for long-term duration, which help for tissues oxygenation and nourishment. This local anesthetic agent might suppress the ectopic discharges from injured neurons, thus slowing nociceptive transmission [21,25,26,27,28]. More time is necessary for improving the clinical symptoms, either pain and radiculopathy or myelopathy, in drop foot patients. According to a recent study that involved 20 patients with upper LDH indicated statistically significant improvement in pain and myelopathy as a mean follow-up period of 13 ± 2.5 months postdecompression and after fixation operation [13].

## 5. Conclusions

The upper LDH with lesion between vertebral column on Th11 to L3 might have serious complications to the lowest part of cord (epiconus and conus medullary), which is parallel to L2 and L3 levels. Drop foot is one of the clinical findings that represented quiet of severe neurologic deficits. Rapid treatment for dropping foot symptoms due to herniated discs must be a priority as the delayed time might cause decrease improvement possibility of approximately 33%. The lateral and posterolateral disc herniation have better improvement than posterocentral herniation or patient with spinal canal stenosis. The transforaminal approach of discectomy followed by CESI under fluoroscopic guidance is a safer and minimally invasive treatment with promising outcomes. The well-trained or proficient operators become an absolute requirement because the accuracy of determining the discectomy site depends on their ability to measure in gentle movement. Chronicity of compression leads to altered local circulation, leading to problems with better oxygenation or nourishment that play a role in nerve recovery. Long-term duration of weakness, preoperative muscle strength, direction and stages of herniated disc, and single or multiple sites of lesions could be the significant prognosis factors. The upper lumbar showed worse lesions than lower lesions associated to drop foot outcomes postoperatively. Further postoperative monitoring should be done by the administration of neurotrophic drugs. It is based on the possibility of neurotrophic drugs for promoting spinal nerve recovery.

The authors acknowledge that there is still a limitation in this case report, such as postprocedure MRI examination, which was difficult to do. Because of the government regulation for restricted traveling during this COVID-19 era and as patients’ residents in suburban areas were not in the same city as Dr. Kariadi Hospital, they have been referred from other hospitals due to the limitation of skilled physicians, despite managing medication and physical therapies priorly. Not every hospital has MRI facilities, so does the hospital close to where they live.

## Figures and Tables

**Figure 1 brainsci-10-00539-f001:**
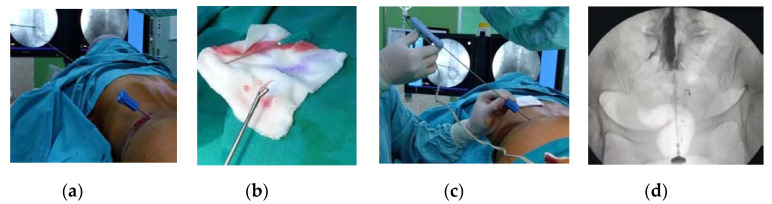
Percutaneous discectomy under C-arm fluoroscopy guiding (**a**) disc tube placement on one-third posteriorly of the discs, (**b**) disc material removal, (**c**) bipolar coagulation, and (**d**) caudal epidural steroid injection (CESI) procedure.

**Figure 2 brainsci-10-00539-f002:**
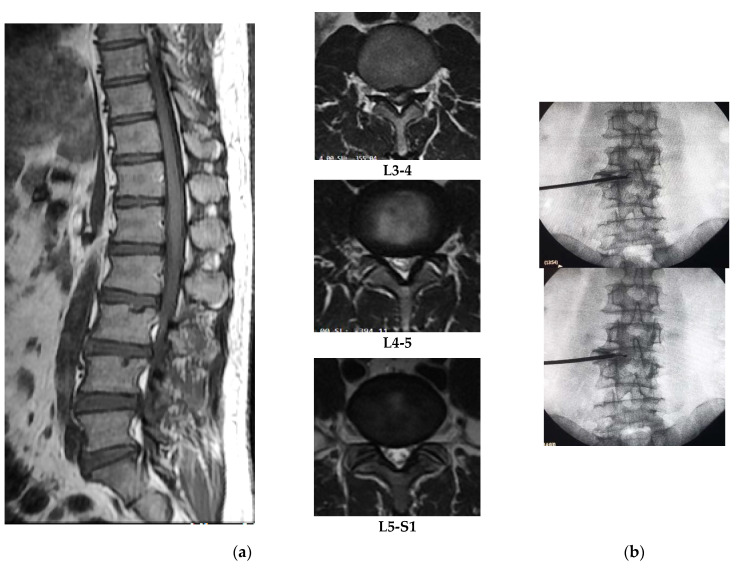
(**a**) Magnetic resonance imaging (MRI) result of patient #1: posterocentrally protruded disc at L3-4, bulged discs at L4-5 and L5-S1 level and the compression slightly to the left side by axial view; (**b**) according to the lesion site dilator canula placed near to the L3-4 neuroforamen (one-third portion between midline and lateral border of disc) by posteroanterior (PA) view, so discectomy and bipolar coagulation might be done carefully.

**Figure 3 brainsci-10-00539-f003:**
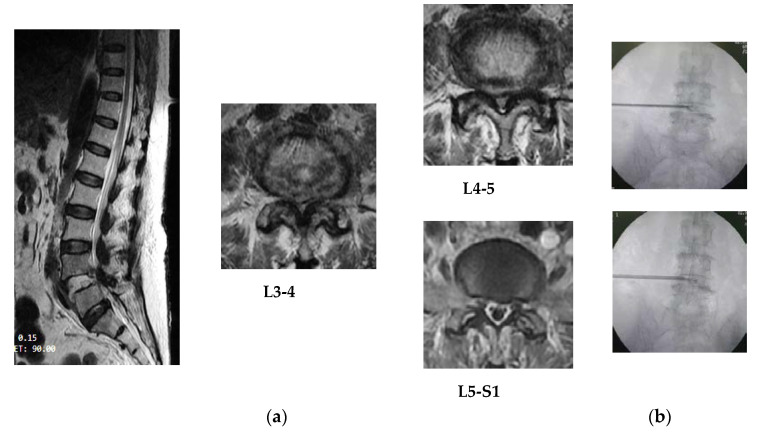
(**a**) MRI result of patient #2: posterocentrally bulged discs at L3-4, L4-5, L5-S1 by lateral view and protruded discs at L4-5 with severe lateral stenosis by axial view; (**b**) according to the site of lesion on midline (posterocentral disc herniation), the dilator canula should be placed on half portion between midline and lateral border of disc by PA view.

**Figure 4 brainsci-10-00539-f004:**
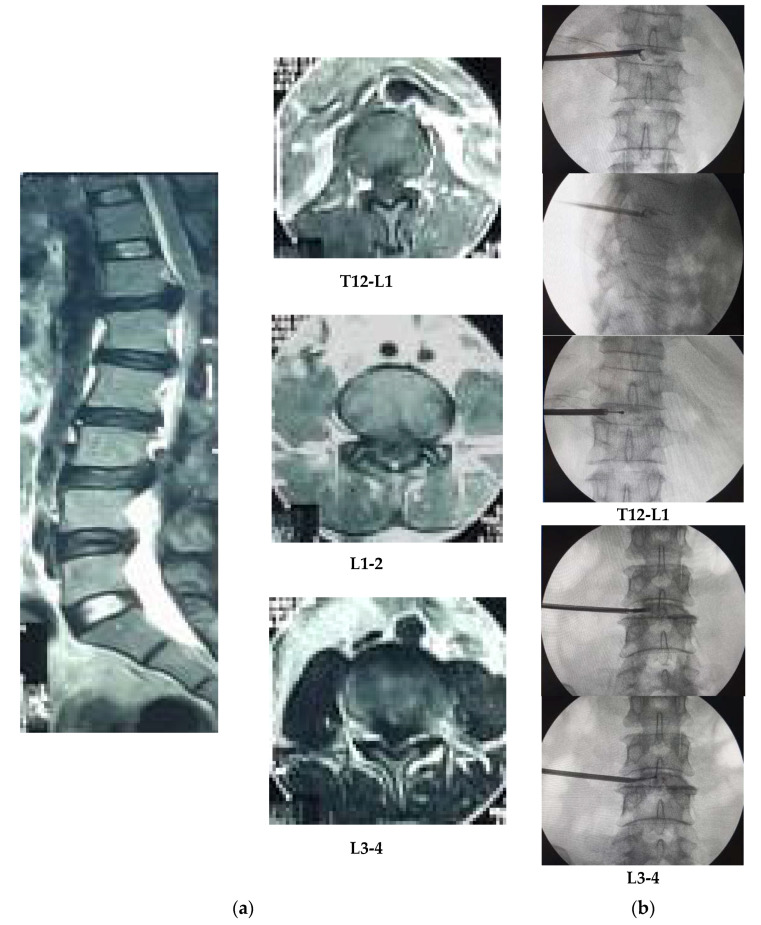
(**a**) MRI result of patient #3 showed extruded disc on T12-L1 and L3-4 levels and protrusion on L1-2 level by lateral view, posterolaterally protrusion on L1-2 and L3-4 levels and posterocentral compression on T12-L1 level and slightly to the left side by axial view; (**b**) according to the site of lesion that went to posterocentral and slightly to left side (T12-L1 and L3-4), the dilator canula and discectomy, either bipolar coagulation, had not far to the midline (one-fourth portion between midline and lateral border of disc) by PA view.

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
