# Peer review of "Percutaneous Discectomy Followed by CESI Might Improve Neurological Disorder of Drop Foot Patients Due to Chronic LDH"

_brainsci, 2020, doi:10.3390/brainsci10080539_

Round 1
Reviewer 1 Report
A case report entitled "Percutaneous Discectomy Followed by CESI Under Fluoroscopic Guiding Might Help for Improvement of Drop Foot Patients Due to Chronic LDH" by Budisulistyo and Firmansyah.
There are some concern and suggestion to improve the case report as
- Re-write the abstract for better understanding and reading.
- Write the full form of each abbreviated word when the first time mention.
- The authors did not mention patient consent for the study.
- The authors should present MRI images of all three patients pre and post-6 months evaluation side by side comparison.
- What is the status of patients' pain levels? Provide the data for the same.
- Authors need to correct some writing issues such as space before unit.
Author Response
- Re-write the abstract for better understanding and reading: Yes. I will re-write it in better so understandable sentences.
- Write the full form of each abbreviated word when the first time mention: Yes. Thank you
- The authors did not mention patient consent for the study: Yes. I have forgot to put its in the article. I wil correct it in the article
- The authors should present MRI images of all three patients pre and post-6 months evaluation side by side comparison: There are few difficulties for asking them to do re-examination of MRI because: 1) in Indonesia there are not many hospitals have MRI fascilities, only in capital city or big private hospital as the actually all subjects are National Health Insurance which need steps priorly until reach to capital city, 2)their living address are in suburban area not in the same city with authors and travel restrictions regulation related to Covid-19 states here
- What is the status of patients' pain levels? Provide the data for the same: Yes. I will correct it in the article
- Authors need to correct some writing issues such as space before unit: Yes. I will correct it
Reviewer 2 Report
This manuscript needs extensive editing of English language and style.
My other concern is that the illustrated cases probably do not add up to the neurology mentioned in the cases; the team is probably concentrating more on the scans than the clinical co-relation when deciding the levels to do. Need to elaborate the rationale.
Author Response
Yes. Thank you.
Will correct it properly
Round 2
Reviewer 1 Report
- Make abstract clear and running by removing subheadings such as Introduction, Case Report, Discussion, and Conclusion.
- Replace y.o. with year or years as per the situation.
- Apart from everything else you need lots of editing needed for this manuscript before publication
Author Response
Response to Reviewer 1 Comments
Point 1: Re-write the abstract for better understanding and reading
Abstract: (1) Introduction: Drop foot caused by mechanical compression due to LDH is a serious problem associate to epiconus and conus medullary syndromes. (2) Case Report: Three patients have suffered from drop feet, numbness, bowel, and bladder problems due to chronic compression of LDH. Patient #1 is a male (35 years old, BMI= 23.9), point 1 on MMT, with protrusion on L3 to S1 discs; Patient #2 is a female (62 years old, BMI= 22.4), point 3 on MMT, with protrusion, on L2-4 and L5-S1discs; Patient #3 is a female (43 years old, BMI= 26.6), point 4 on MMT, with extrusion on T12-L1 and L1-2 and L3-4 protruded discs. The sixth-month evaluation showed: improvement of MMT scale, stand, and walkability with Patient #1 and #2. Patient #3 showed improvement in bowel and bladder problems within ten weeks afterward. All have no any suffering complains of pain.(3) Discussion: Patient #1 and #2 showed better clinical outcomes than Patient #3 who suffered from epiconus and cauda equina syndromes. Triamcinolone and lidocaine act as analgesic and anti-inflammation which improvement intraepidural circulation adjacent to the lesion sites. Conclusion: Drop foot that is caused by mechanical compression of LDH ought to be treated immediately. Lateral or posterolateral compression has better outcomes associated with the affected anatomical structures. The transforaminal approach of discectomy followed by CESI under fluoroscopic guidance is safer because of the minimally invasive treatment with promising outcomes.
Keywords: LDH; drop foot; fluoroscopic; discectomy; CESI
Point 2: Write the full form of each abbreviated word when the first time mention
Abbreviations: LDH: lumbar disc(s) herniation, MMT: manual muscle test, BMI: basal metabolism index, CESI: caudal epidural steroid injection, LBP: low(er) back pain, PA: postero-anterior, AP: antero-posterior, TFMD: transforaminal micro discectomy, NRS: numerical rating scale, MRI: magnetic resonance imaging,, PTR: patellar tendon reflex, ATR: ankle tendon reflex, PELD: percutaneous endoscopic lumbar discectomy, CTM: computerized tomo-myelography, LOS: length of stay, ESI: epidural steroid injection
Point 3: The authors did not mention patient consent for the study
In my first submitted article have not written as well, so I put their informed consents on subtitle: Procedure Technique
“…All patients have been informed of the purpose, benefits, and potential complications or the side effect and signed the informed consent. The potential complications are: worsening of the motor, sensory or bowel, and bladder, persistentpain, hypersensitive reaction regard to local anesthetic (lidocaine), dye contrast agent, antibiotic, the discectomy kits or suture material. Percutaneous discectomy procedure only requires a local anesthetic administration, so the patients are still awake and able to communicate with the operator. When the needle or dissection procedures touch the nerve root or other inconveniences to minimize the potential complication. Moreover, during the procedure, the operator can check the patient’s motor or sensory function.”
Point 4:
Actually there are few difficulties for asking MRI re-examination cause of:
- There are not many hospitals in Indonesia belongs to MRI fascilities, only in capital or big private hospital. Actually all subjects were covered by national health insurance which need steps priorly until reach to capital hospital.
- Subjects living addresses are in suburban areas and one of them in another island, while local authorizations have been running for travel restriction regard to covid-19 status here.
Point 5: What is the status of patients pain levels?
- Post operative monitoring of pain complaints commonly in moderate level (NRS = 4-6) for 1-2 weeks after surgery. It gradually reduce to mild level (NRS = 2-3) and subjects still prescribed of analgesics medication in small doses.
- “They felt of mild pain on the level sites but can treated with gabapentin (150-200 mg/ day) and acetaminophen (1000-1500 mg/ day). Thus acetaminophen is only given when the pain gonna be moderate level, despite consuming gabapentin regularly. Another medication drugs such as thiamine (100 mg/ day) and calcium lactate (1000 mg/ day) still taken regularly.“
Point 6:
- I have correct it such as: 1 mL, 1.5 mg, etc